# Vortex Fluidic Mediated Oxidative Sulfitolysis of Oxytocin

**DOI:** 10.3390/molecules27031109

**Published:** 2022-02-07

**Authors:** Emily M. Crawley, Scott Pye, Briony E. Forbes, Colin L. Raston

**Affiliations:** 1Flinders Institute for Nanoscale Science and Technology, College of Science and Engineering, Flinders University, Adelaide, SA 5042, Australia; emily.crawley@flinders.edu.au (E.M.C.); scott.pye@flinders.edu.au (S.P.); 2Flinders Health and Medical Research Institute, College of Medicine and Public Health, Flinders University, Adelaide, SA 5042, Australia; briony.forbes@flinders.edu.au

**Keywords:** vortex fludics, sulfitolysis, cysteine, peptides

## Abstract

In peptide production, oxidative sulfitolysis can be used to protect the cysteine residues during purification, and the introduction of a negative charge aids solubility. Subsequent controlled reduction aids in ensuring correct disulfide bridging. In vivo, these problems are overcome through interaction with chaperones. Here, a versatile peptide production process has been developed using an angled vortex fluidic device (VFD), which expands the viable pH range of oxidative sulfitolysis from pH 10.5 under batch conditions, to full conversion within 20 min at pH 9–10.5 utilising the VFD. VFD processing gave 10-fold greater conversion than using traditional batch processing, which has potential in many applications of the sulfitolysis reaction.

## 1. Introduction

Oxidative sulfitolysis introduces negatively charged sulfonate moieties onto a peptide chain containing cysteine residues [1,2]. This increases the solubility and stability of proteins, and is especially useful for processing proteins with multiple cysteine residues [3,4,5]. Sulfitolysis is reversible, specific to disulfide moieties [6,7,8], and can improve the folding of proteins [9,10,11]. A distinct advantage of sulfitolysis is the formation of an intermediate S-sulfonate, which is essentially a protecting group during the protein modification, and can be readily removed by treatment using excess thiol when required [2]. The presence of small amounts of a reducing agent, such as β-mercaptoethanol or dithiothreitol, during the removal of this protecting group by oxidation stimulates disulphide bond reshuffling [12]. The oxidative sulfitolysis of cysteine residues in proteins involves treatment with sodium sulphite and an oxidizing agent, typically sodium tetrathionate or potassium *o*-iodobenzoate. Sulfitolysis was first reported more than 50 years ago [6], but the reaction is poorly understood. In addition, it is also noteworthy that the by-products in the production of many biologically active peptides and proteins result in inactive or less active precursors, which must undergo processing to obtain their full potency [1]. The use of a sulfitolysis step can enhance the efficiency of this processing to achieve the properly folded protein with the desired activity, efficacy, and affinity levels [13].

Cysteine containing proteins expressed in *E. coli* cells as inclusion bodies benefit from oxidative sulfitolysis following inclusion body solubilisation to increase the yield of refolded protein [10]. Oxidative sulfitolysis can be used to cleave all cysteine linkages, thereby protecting the thiols as sulphites (Figure 1). Such protection allows for greater tolerance for purifying the protein, which can then be easily reduced back to the disulphide of the correctly folded protein. Oxidative sulfitolysis has been widely used in biotechnology for the isolation and analysis of cysteine-containing proteins [1,14].

Oxidative sulfitolysis has been used to separate the two chains of insulin [15,16], as well as in the production of the proinsulin precursor using *E. coli* expression systems [17,18]. When proinsulin is produced in bacteria, it typically forms misfolded proinsulin as inclusion bodies. Human insulin contains three disulfide bonds that are essential for its native conformation, and the formation of the native disulphide bonds is the rate-determining step during insulin folding [19]. Introducing eight S-sulfonate moieties into synthetic hepcidin was shown to considerably decrease aggregation and, under optimised conditions, dramatically increase the yield of refolded protein [4]. Oxidative folding of synthetic hepcidin is inherently difficult due to its high cysteine content (≈30%) and high aggregation propensity [20,21].

Oxytocin is a small peptide hormone and neuropeptide secreted in humans by hypothalamus nerve cells. It is stored in the hypophysis posterior lobe [22] and is well-known for its role in lactation and parturition [23]. The nine amino acid sequence of oxytocin (Cys-Tyr-Ile-Gln-Asn-Cys-Pro-Leu-GlyNH_2_) was elucidated in 1953 [24,25], having an intramolecular di-sulfide bridge between two cysteine moieties (Figure 1). Production of oxytocin is limited, due to difficulties in its isolation and purification. The chemical synthesis of oxytocin is laborious and requires purification to remove the impurities which have chemical properties close to that of oxytocin [22]. In addition, the bacterial approach to prepare oxytocin, using recombinant DNA techniques, has shown promise, but short peptides have small lifetimes in bacterial cells [22].

One example of oxytocin production utilises sulfitolysis of a fusion protein expressed from *E. coli* [22]. This hybrid protein included the oxytocinoyl lysine tetramer and a histidine hexamer in the C-terminus of the molecule. The use of the histidine hexamer enhances the solubility of the molecule as well as simplifying the purification process, with the presence of the oxytocinoyl lysine tetramer increasing the final yield of oxytocin. The hybrid protein was expressed as insoluble inclusion bodies, which then required solubilisation using urea and reducing agent, namely dithiothreitol (DTT), affording the completely reduced hybrid protein, which only exists in this form for a short period, with the high concentration of urea hindering purification. In contrast, sulfitolysis allowed the complete reduction of the disulphide bonds with an increase in solubility of the sulfonated product and prevention of aggregation, even in the absence of urea [22].

The vortex fluidic device (VFD), Figure 2, is effective in controlling oxidation processes [26], and we hypothesised that the high shear stress and associated mass transfer in the device would be effective in mediating the oxidative sulfitolysis reaction of oxytocin, in producing the S-sulfonate analogue. The VFD is a thin film microfluidic platform with a diversity of applications, including in controlling organic reactions [26,27], promoting biochemical transformation [28,29], mediating materials chemistry [30], and more. Protein folding in pharmaceutical and agricultural industries rely heavily upon laborious processing methods. The VFD is effective in refolding proteins, as established for hen egg white lysozyme (HEWL) and cAMP-dependant protein kinase A (PKA), with the processing requiring much smaller solution volumes with refolding times reduced by a factor of 100 compared to the commonly used overnight dialysis [28].

The dynamic thin film in a VFD, Figure 2, has high shear topological fluid flow regimes of submicron dimensions, depending on the properties of the liquid and rotational speed, at a tilt angle, θ of 45°, which has proven optimal for all applications of the device [31]. The topological fluid flows are (i) a spinning top shape at low rotation speed in water (above ca. 3.5 k rpm), arising from the Coriolis force from the hemispherical shape of the base of the tube, and (ii) double helical flow arising from the eddies from Faraday waves across the thin film being twisted by the Coriolis force from the curved surface of the wall of the tube [31]. There is also a special case of spherical or spicular flow where (i) and (ii) have the same diameter [31]. Processing in the VFD can occur under continuous flow where jet feeds deliver reagents at points inside the tube, or in the so-called confined mode, where a specific volume of liquid is added to the angled tube and then it is spun in the same way. Confined mode features in the present study, given the small volumes to be processed and that the processing time as such is longer than the residence time for liquid passing through the VFD, even for flow rates less than 0.5 mL·min^−1^ [31]. Nevertheless, with the above detailed understanding of the fluid flow in the VFD, translating any process from confined mode to continuous flow is well established [31]. A standard 20 mm OD quartz tube (17.5 mm ID), 19 cm in length with a hemispherical base, was used for all processing herein.

## 2. Materials and Methods

Oxidative sulfitolysis was performed on oxytocin using the following procedure. First, a series of carbonate buffered solutions were prepared by the dissolution of NaHCO_3_ (168 mg, 2 mmol) and Na_2_CO_3_ (21.2 mg, 0.2 mmol) in ultrapure water (20 mL). The pH of each of these solutions was raised with NaHCO_3_ or lowered with Na_2_CO_3_ to produce a series of pH solutions of 9, 9.5, 10, and 10.5. Next, the stock solution of oxytocin was prepared by dissolving oxytocin (1 mg, 9.93 × 10^−7^ mol, 0.1 mM) in each of the carbonate buffered solutions (10 mL). A stock solution containing both Na_2_SO_3_ and Na_2_S_4_O_6_ was then prepared by dissolving Na_2_SO_3_ (100 mg, 0.8 mmol, 80 mM) and Na_2_S_4_O_6_ (5.4 mg, 0.02 mmol, 20 mM) in each of the carbonate buffered solutions (10 mL).

The reactions were then performed in either a magnetically stirred vial or operated in the VFD at either 4.5, 7, or 9 k rpm. For these, the oxytocin stock (800 µL) and the Na_2_SO_3_–Na_2_S_4_O_6_ stock (80 µL) were combined and diluted in the pH buffered carbonate solution (3120 µL) and reacted for a total of 20 min at room temperature. The resulting reaction mixture contained oxytocin (0.02 mM), Na_2_SO_3_ (1.6 mM), and Na_2_S_4_O_6_ (0.04 mM). Upon completion of the desired time, the reaction was quenched by the addition of trifluoro-acetic acid (20 µL).

Aliquots (110 µL) were then centrifuged (1 min at 10,000× *g*) for HPLC analysis. Aliquots (500 µL) were also diluted 1:1 with water for HPLC-MS analysis. This was performed to identify the peaks in the HPLC trace, whilst a UV detector at 215 nm was used for determining peak integrals for the conversions. This was repeated in triplicate, with results shown in Figure 3 and Figure 4. All HPLC was performed on an Agilent Prep-C_18_ column (10 µm, 4.6 × 250 mm, 100 Å) connected to the Agilent 1260 Infinity Quarternary LC system. Buffer A: 0.1% aqueous trifluoracetic acid. Buffer B: 80% acetonitrile in 0.08% trifluoracetic acid. Peptides were eluted using a linear gradient of 10–50% acetonitrile for 40 min, at a flow rate of 0.5 mL/min. UV detection was at 215 nm and peak areas were used for quantitation of peptides. T_R(Oxytocin)_ = 28.76 min and T_R(Sulfitolysed Oxytocin)_ = 24.67 min (Figure 5).

LC-MS was conducted using the Waters Synapt HDMS with LC-MS/MS capability using Acquity UPLC. A C18 column was used with flow rates of 0.2 mL.min^−1^. A binary solvent system was used, the solvents being a formic acid solution (0.1% *v*/*v* aq.) and acetonitrile. Initial flow consisted of 90:10 formic acid: acetonitrile, before ramping to 70:30 over 10 min. This ratio was held for 4 min giving a total time of 14 min. Mass detection was 300–2000 *m*/*z*. A photodetector at 252 nm was also used (Appendix A).

Statistical analysis of conversions to sulfonated products using different reaction conditions or processing were performed using a 2-way ANOVA with post-hoc Tukey’s multiple comparisons tests. Significance is reported as ns (not significant), * (*p* < 0.01), ** (*p* < 0.001), or *** (*p* < 0.0001). All statistics were performed using GraphPad Prism v8.

## 3. Results and Discussion

Oxidative sulfitolysis of oxytocin was performed using carbonate buffered solutions (1 mL) in a series of pH solutions of 9, 9.5, 10, and 10.5. For each pH, oxytocin (0.02 mM), sodium sulphite (1.6 mM), and sodium tetrathionate (0.04 mM) were processed for 20 min in either a magnetically stirred vial (designated herein as batch processing), or in the VFD at rotational speeds of either 4.5, 7, or 9 k rpm. These speeds were chosen to span the different complex fluid dynamics in the device for increasing rotational speeds, with 4.5 k rpm in water dominated by spinning top topological flow and 9 k rpm dominated by double helical flow, Figure 2 [31]. Reactions were quenched by the addition of trifluoro-acetic acid (20 µL) [2,15,32]. The analysis of the products was performed by HPLC (Figure 5 and Appendix A) and LC-MS (Appendix A) to confirm the identify of peaks in the HPLC trace. All reaction conditions were repeated in triplicate with significance reported as ns (not significant), * (*p* < 0.01), ** (*p* < 0.001), or *** (*p* < 0.0001).

The oxidative sulfitolysis performed on oxytocin under normal batch conditions with increasing pH shows that the optimal pH is 10.5 for 20-min reaction times (Figure 3). At pH 9, 9.5, and 10, conversion to the S-sulfonate product is <15%, whereas at pH 10.5, the conversion is >99%. This establishes that the oxidative sulfitolysis of oxytocin in a carbonate buffer is dependent on pH, and that the reaction will not achieve completion in the 20 min investigated here when the pH < 10.5.

However, when processed in the VFD, full conversion can be achieved throughout the entire buffer range of carbonate, from pH 9 to 10.5, with the highest rotational speed, 9 k rpm, being effective for near full conversion at the lowest pH of 9 (Figure 4). This result is yet another example of the unique capabilities of the VFD and is in accord with the high mass transfer and heat transfer in the thin film. Where the spinning top and double helical topological fluid flows strike the surface of the tube, high temperatures can be achieved, in excess of 270 °C, as established by being able to melt elemental bismuth in an organic solvent in the device [31]. Such high temperatures in localized regimes on the surface of the tube can increase the dissociation of water, although the time scale for being exposed to such forcing conditions is small [31]. In this context, the ionic product of water is 1000-fold greater at 240 °C than at 25 °C, i.e., at high temperatures water is a stronger acid and base [33]. Recent examples utilizing the high shear in the VFD as such include surfactant-free fabrication of fullerene C_60_ nanotubes [34], transformation of graphite into highly conducting graphene scrolls [35], direct transesterification of wet microalgae biomass to biodiesel [36], and synthesis of macroporous bovine serum albumin-based microspheres [37].

Oxidative sulfitolysis performed under conventional batch and VFD processing (confined mode), for the same pH range shows a statistically significant enhancement in the VFD at pH 9, 9.5, and 10. Thus there is an increase in the reaction rate for the oxidative sulfitolysis performed under VFD processing. This suggests that where the sulfitolysis reaction is limited by a pH range below 10.5, it may be viable using VFD processing rather than conventional batch processing. The clear advantage of the VFD in the present study relates to the high mass transfer under high shear, with the chemistry not limited to diffusion control [38]. Examples of the processing resulting in oxidative sulfitolysis of oxytocin and other proteins have timescales on the order of hours to days, Table 1 [1,4,10,22,39,40,41]. The utilisation of the VFD, affording >95% conversion at a range of rotational speeds and different pH values, are all within 20 min.

## 4. Conclusions

This widely used reaction in protein production and purification gave >10-fold conversion using VFD processing than previously reported using traditional batch methods. This is significant given that the extent of recovery of biological activity is important in many applications of the sulfitolysis reaction.

The 20 min processing required in the VFD circumvents the use of the VFD under continuous flow where the reaction time is much less, as highlighted above. However, we envisage the use of the VFD with robotic control for adding a specific volume of liquid for processing, then removing, and adding another aliquot of the feedstock for processing in the VFD, and so on and forth, has potential in adopting VFD processing for the sulfitolysis reaction. Employing the VFD to allow access to improvements in reactions limited by a pH range could be expanded to a wide range of applications for research where the localised in situ dissociation of water under shear stress drives the chemistry.

## Data Availability

The data used to support the findings of this study are available from the corresponding author upon request.

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
