# Peer review of "Vortex Fluidic Mediated Oxidative Sulfitolysis of Oxytocin"

_molecules, 2022, doi:10.3390/molecules27031109_

Round 1

Reviewer 1 Report

The manuscript by Crawley et al., demonstrates the use of an angled vortex fluidic device (VFD) for oxidative sulfitolysis of oxytocin.

The introduction part is well written and the context is very clearly stated. Also the results and discussion part is very clear, interesting and well written.

At least to me, the manuscript is acceptable for publication provided that some very minor alterations are made.

1) please be consistent either use x/y or x y-1 (line 101: 0.5 mL. min-1. while on line 195 mL/min.)

2) line 194: LC-MS/MS not LCMS/MS

3) Maybe it also worth mentioning how many times you have repeated an experiment (triplicate,…) in the caption of some figure.

Otherwise, interesting work!

Author Response

All comments addressed in full.

Reviewer 2 Report

The authors describe a method of processing small peptides with cysteine residues. The results presented will be of particular interest to scientists with an interest in bulk processing of peptides.The data is clearly represented and convincing. This reviewer recommends publication.

Author Response

No issues to address.

Reviewer 3 Report

Crawley et al manuscript entitled "Vortex Fluidic Mediated Oxidative Sulfitolysis of Oxytocin" is an interesting article.

In this article, the authors conducted interesting experiments in which they developed a versatile process exploiting VFD to investigate oxidative sulfitolysis at varying pH changes.

The strength of the article is that it assesses the oxidative sulfitolysis of oxytocin using VFD as it yields S-sulfonate analog. In addition, parameters like topological fluid flow, dimension analysis, rotational has been extensively researched and standard quartz tube was selected for all processing. 

However, there is also the weakness of the article as below.
Line 8-12: Abstract: This section needs a few additional sentences.
- Definition of oxidative sulfitolysis for the opening sentence may look better along with the previous and current state of knowledge in the field (Example: line 16-17 and line 28-30).

-Line 12: Please provide the summary statement in the last sentences of the abstract sections.

Line 33-40:  Please add the perspective for higher animal models (eukaryotic system), if similar reactions are conducted in those higher forms.

Line 166: The heading "Materials and Methods" and their contents should be written after the "Introduction" section. In other words, the "Result and Discussions" section should follow "Materials and Methods".

Line 213: I would suggest replacing "20 min" with more generic language. For example, the shortened timeframe/or longer processing achieved ........ was due to improvements in .....

In summary, the article is well written and I do not have other outstanding issues.

Author Response

 All issues have been addressed, except for ‘Please add the perspective for higher animal models (eukaryotic system), if similar reactions are conducted in those higher forms.’ We are at a loss as to what the reviewer is after, and no action has been taken on this issue.